# Semantic Retrieval of Remote Sensing Images Based on the Bag-of-Words Association Mapping Method

**DOI:** 10.3390/s23135807

**Published:** 2023-06-21

**Authors:** Jingwen Li, Yanting Cai, Xu Gong, Jianwu Jiang, Yanling Lu, Xiaode Meng, Li Zhang

**Affiliations:** 1College of Geomatics and Geoinformation, Guilin University of Technology, Guilin 541004, China; 2Ecological Spatiotemporal Big Data Perception Service Laboratory, Guilin University of Technology, Guilin 541004, China

**Keywords:** remote sensing image retrieval, visual feature word bag, bag-of-words, association rules

## Abstract

With the increasing demand for remote sensing image applications, extracting the required images from a huge set of remote sensing images has become a hot topic. The previous retrieval methods cannot guarantee the efficiency, accuracy, and interpretability in the retrieval process. Therefore, we propose a bag-of-words association mapping method that can explain the semantic derivation process of remote sensing images. The method constructs associations between low-level features and high-level semantics through visual feature word packets. An improved FP-Growth method is proposed to achieve the construction of strong association rules to semantics. A feedback mechanism is established to improve the accuracy of subsequent retrievals by reducing the semantic probability of incorrect retrieval results. The public datasets AID and NWPU-RESISC45 were used to validate these experiments. The experimental results show that the average accuracies of the two datasets reach 87.5% and 90.8%, which are 22.5% and 20.3% higher than VGG16, and 17.6% and 15.6% higher than ResNet18, respectively. The experimental results were able to validate the effectiveness of our proposed method.

## 1. Introduction

As remote sensing image scenes become more and more complex and the amount of information contained in the images becomes more and more difficult for accurate cross-modal retrieval of remote sensing images, how to extract the required images from a large number of remote sensing images using high-level semantics has become a hot topic in remote sensing image processing applications in recent years. It is important to establish the relationship between non-implicit features and high-level semantics to improve the interpretability of the cross-modal retrieval process and to give users the knowledge they need to better understand the retrieval process and find the desired images.

The current stage of research on remote sensing image retrieval is dominated by text-based image retrieval (TBIR) [1] and content-based image retrieval (CBIR) [2,3]. The text-based image retrieval method uses the geographic regions, acquisition time and sensor types as retrieval criteria [4,5]. The reliance on manual annotation of keywords in images makes this task time-consuming and labor-intensive [6]. The content-based retrieval method mainly compares the similarity between the query image and the images in the image retrieval dataset by extracting the features of the im-age, and then arranges the retrieved images according to the similarity as the retrieval results [7,8,9]. The accuracy of the retrieval can be significantly affected by the extracted features. The accuracy of the retrieval depends only on the extracted features. As a result, the low-level features of the extracted images are challenging to characterize with high-level semantics that humans can grasp [10]. Because of this, the majority of work on CBIR has concentrated on extracting various feature descriptors in an effort to mimic human visual perception as closely as possible and hence increase retrieval accuracy.

In order to make up for the shortcomings of the above two image retrieval methods, some scholars began to study the semantic-based remote sensing image re-retrieval method, which obtains the semantic concept of images and uses it as the basis of semantic retrieval [11]. Deep learning techniques can achieve automatic extraction and annotation of semantic information of remote sensing images, which has significant advantages in remote sensing image classification and remote sensing scene recognition, but it is difficult to explain the process of deriving semantics. However, it is difficult to derive high-level semantics directly using the underlying visual features, so a method needs to be constructed that can establish a deep connection between the two [12]. Therefore, data mining techniques can be used to construct the connection between the underlying features of remote sensing images and the high-level semantics, so that the process of deriving semantics can be interpretable [13]. However, there are still two problems to be solved: (1) Remote sensing images are raster data, which cannot be mined directly using association rule algorithms, and they need to be discretized, otherwise sharp boundary problems will occur [14,15]. (2) The association rule mining algorithm also has the problem of mining all rule sets in the data as much as possible, which leads to a large number of invalid rules in association rules [16,17], thus increasing the cost of mining, and the knowledge obtained from the invalid rules cannot express the association relationship between image features and image semantics. This makes its practical application difficult.

In order to achieve semantic-based remote sensing image retrieval and solve the problem of lack of interpretability in the retrieval process, this paper proposes a bag-of-words association mapping method for establishing the connection between the basic features of remote sensing images and high-level semantics. In this method, the visual feature bag-of-words algorithm is used to discretize the features and reduce the dimensionality. The improved FP-Growth algorithm is then used to obtain the association rules. The proposed method can complete semantic retrieval of remote sensing images with excellent retrieval accuracy and quick retrieval time in the publicly accessible dataset AID with NWPU-RESISC45. Our main contributions are as follows:(1)A bag-of-words association mapping method is proposed by which the semantic retrieval of remote sensing images can be realized. In particular, the visual bag-of-words method is used to solve the problems of too high dimensionality and incomplete description of the underlying features of remote sensing images and to discretize the continuous attribute values in image features by combining the similarity between the features. The FP-growth algorithm used in the method is also improved to quickly obtain the effective rule set.(2)Based on the image semantic retrieval implemented using the proposed method, we propose a feedback mechanism by which incorrect retrieval results can be semantically corrected to reduce the probability of incorrect semantics in them in order to improve the accuracy of the next retrieval.

The rest of the paper is organized as follows: Section 2 presents work related to image semantic retrieval and image retrieval feedback. Section 3 presents the proposed approach and describes the process of linking high-level semantics with underlying features for semantic retrieval of remotely sensed images. Section 4 describes the dataset and evaluation metrics used and the designs associated with the experiments. Section 5 discusses the impact of the parameters involved in the method of this paper on retrieval. Section 6 concludes the paper.

## 2. Related Works

This section reviews the literature relevant to the work in this paper, including recent research on semantic retrieval methods for remote sensing images with interpretability, and image retrieval feedback methods for correcting retrieval results.

### 2.1. Image Semantic Retrieval

Implementing semantic retrieval of remote sensing images allows users to retrieve similar images and plays an important role in many practical applications. The key to accomplishing this is to establish associations between the underlying features of the image and the high-level semantics. Association relationships are established using association rule mining (ARM) methods, which can extract potential rules from large amounts of data and provide hidden value services [18]. Methods based on convolutional network frameworks are often used to construct such connections. For example, a new Deep Hash Convolutional Neural Network (DHCNN) is used to retrieve similar images and classify the semantic labels of images for visual and semantic retrieval of images [19]. With consideration of the semantic complexity of the scene that remote sensing images have, the trained FCN is used to predict the segmentation map of remote sensing images. Then the obtained multi-label vector is employed to extract the regional convolutional features of the segmentation map based on the region-based multi-label retrieval [20]. Besides using the convolutional network framework to establish connections, the rules mined by association rule algorithms in data mining techniques can also accomplish the conversion from the underlying features to the higher-level semantics of remotely sensed images. Some scholars have extended this idea and applied it to image retrieval, such as for multimodal semantic association rule-based approaches that can fuse keywords and visual features for web image retrieval [21]. In addition, effective image retrieval can also be achieved through the use of various data mining techniques. Raniah A et al. combined clustering and association rules to achieve multimodal retrieval of images [22]. However, this approach does not sufficiently take into account the effect of probabilities carried in strong rules on the retrieval results. In remote sensing images, similar or identical spatial distribution features can be found for similar objects. Association rules are used to better mine the spatial distribution rules among pixels and implement semantic remote sensing image retrieval [23].

### 2.2. Image Retrieval Feedback

To increase the precision of remote sensing image retrieval, retrieval feedback approaches are crucial [24]. The basic principle of the feedback technique is to correct the initially returned retrieval ranking [25]. The method includes pseudo-relevant feedback and display-relevant feedback. Pseudo-relevance feedback is automatic feedback. It can take the top retrieval results in the retrieval list as relevant images and uses their features as the basis for query expansion. Feedback on display relevance is completed manually. Users must identify images in each search ranking that do not meet the search criteria and then rerank them using a re-weighting approach [26], a probability distribution-based approach [27], or a machine learning-based approach [28]. Kundu proposed a new CBIR retrieval method and used a graph-theory approach to rank images based on user feedbacks to improve retrieval performance [29]. M. Schroder extended the steps of Bayesian inference and focused on the interactivity of the proposed method to provide immediate feedback in the form of instantly updated a posteriori maps [30]. An SVM-related feedback CBIR algorithm based on feature reconstruction was proposed and validated on a large dataset by Wang [6]. Hamed Qazanfari proposed a correlation feedback-based learning method based on the near- or far-parent model to address the problem of features not accurately representing image content and evaluated it based on the dataset Corel-10k [31].

### 2.3. Related Dataset

There are many remote sensing image datasets that are used to evaluate retrieval and classification performance, such as the UC Merced dataset (UCMD), the WHU-RS19 dataset, the Aerial image dataset (AID), the RSSCN7 dataset, the NWPU-RESISC45 dataset (NWPU45), and the PatternNet dataset. The UCMD dataset was manually extracted from large images down-loaded from the United States Geological Survey (USGS), selected from aerial orthophotos with a pixel resolution of 30 cm, and consists of 21 land-use/land-cove (LULC) categories with 100 images in each category. Image retrieval studies using locally invariant features on this dataset outperformed standard features such as color and texture [32]. The WHU-RS19 dataset was collected from Google Earth and has 19 categories with about 50 images each. It is characterized by a large variation in resolution, scale, orientation and illumination of the images [33]. The AID dataset is a large-scale dataset for aerial scene classification, with 10 classes; each class consists of 220–420 images. It is characterized by images from different remote sensing sensors and a large variation in spatial resolution [34]. The RSSCN7 dataset is derived from Google Earth images sampled at four different scale levels and includes seven categories of 400 images each. It is characterized by a large diversity of images of scenes taken under changing seasons and weather conditions [35]. The NWPU45 dataset was constructed by selecting representative classes of scenes and contains 45 classes with 700 chapters of images each, with spatial resolutions ranging from approximately 0.2 to 30 m. It is characterized by large differences in images in terms of translation, spatial resolution, viewpoint, object pose, illumination, background, and occlusion [36]. Weixun Zhou proposed the “PatternNet” a sizable remote sensing dataset that consists of 38 classes with 800 images each. Deep learning-based retrieval techniques are ideal for it because of its size [37].

## 3. General Scheme for Semantic Retrieval of Remote Sensing Images

The goal of this paper is to achieve semantic retrieval of remotely sensed images by entering search terms. Therefore, we have designed a semantic mapping method for remotely sensed images that converts low-level visual features in the image into high-level semantics. Figure 1 outlines the retrieval process accomplished using the methods in this paper.

The retrieval method is divided into three steps: (1) The remote sensing image is cut twice, and feature extraction is performed on the small blocks after the cut to obtain a high-dimensional feature matrix. (2) The visual feature word bag is aggregated through the feature matrix of the training set, the image word set is assembled together with its corresponding semantic set to construct the associated text, and the words-to-semantics mapping rules are obtained according to the improved TP-Growth algorithm. (3) The retrieval library, the remote sensing images complete the process from feature extraction to forming semantic expressions, and calculate the similarity between the semantic retrieval words input by the user and the semantics of the remote sensing images. The remote sensing image with higher semantic similarity is used as the retrieval result, and for the wrong image in the retrieval result, a feedback mechanism is used to correct it in order to improve the retrieval accuracy in the next retrieval.

### 3.1. Remote Sensing Image Feature Extraction

Remote sensing image feature extraction is the basis for semantic retrieval. Firstly, the image is sliced twice to improve the efficiency of retrieving small objects in the image. Then, the color and texture features are extracted from the sliced small blocks and the feature matrix of the block is constructed.

#### 3.1.1. Remote Sensing Image Slice

In the image slicing, the spatial dimension of the remotely sensed image is (w, h) and the slicing ratio is (seg×seg).The specific slicing method is shown in Figure 2.

To better obtain locally relevant information about the image, we perform twice slices of the remotely sensed image. The first slice cut the original image into ${\mathrm{seg}}_{1}{}^{2}$ small target blocks to obtain multi-semantic information. The second slice is to cut each small target block into ${\mathrm{seg}}_{2}{}^{2}$ blocks, with the aim of obtaining its local features for the subsequent construction of correlated text.

#### 3.1.2. Remote Sensing Image Feature Extraction

After completing the second cut, feature extraction is performed on all blocks in each remotely sensed image. Two potential visual features, texture and color, were selected to characterize the remotely sensed images.

Texture feature extraction

A local binary pattern (LBP) [38] was selected to extract the texture features.

The principle is to use a 3 × 3 window to traverse the remote sensing image to extract features and to compare each feature with the pixel values of its neighbors, using the central pixel as a threshold. If the neighboring pixel value is not less than the threshold, it is marked as 1, otherwise it is marked as 0. In this way, by arranging the values clockwise from the top left corner of the window, a set of 8 binary digits can be obtained, which can subsequently be converted to the decimal system. The decimal digits are used as a unit of texture between 0 and 255. A new image is obtained after the traversal is completed, which is a texture image. As shown in Equation (1),
(1)$$\mathrm{LBP}=\sum_{p=0}^{7}s(g_{p}-g_{c})2^{p}$$
where $g_{p}$ is the neighborhood pixel value and $g_{c}$ is the central threshold, while the function s(x) is calculated as shown in Equation (2):(2)$$s(x)=\left\{\begin{array}{l} 1,x\geq 0\\ 0,x<0 \end{array}\right.$$

2.Color feature extraction

In general, the color distribution of an image [39] can be expressed in terms of the first, second and third color moments of the color, as shown in Equations (3)–(5),
(3)$$E_{i}=\frac{1}{N}\sum_{j=1}^{N}p_{i,j}$$
(4)$$\sigma_{i}=(\frac{1}{N}\sum_{j=1}^{N}{\left(p_{i,j}-E_{i}\right)}^{2}{)}^{\frac{1}{2}}$$
(5)$$s_{i}=(\frac{1}{N}\sum_{j=1}^{N}{\left(p_{i,j}-E_{i}\right)}^{3}{)}^{\frac{1}{3}}$$
where, $p_{i,j}$ denotes the *j* image pixel values under the *i* color channel in the remote sensing image, and *N* is the total number of pixels in the image. $E_{i}$ is the first color moment (mean) of the image that indicates the degree of lightness and darkness of the image, $s_{i}$ is the second color moment (standard deviation) of the image that indicates the image color distribution, and $s_{i}$ is the third color moment(variance) of the image that represents the symmetry of the image color distribution.

3.Feature vector set acquisition

The extracted features are arranged sequentially, and the texture features and color features are combined into a feature vector *Feature* as shown in Figure 3.

Feature extraction is performed on all blocks of each remote sensing image to form the feature vector set Features, as shown in Equation (6),
(6)$$\mathrm{Features}=\left[\left({\mathrm{texture}}_{1},{\mathrm{color}}_{1})\;,({\mathrm{texture}}_{2},{\mathrm{color}}_{2}),\ldots ,({\mathrm{texture}}_{{\left({\mathrm{seg}}_{1}*{\mathrm{seg}}_{2}\right)}^{{}_{{}^{2}}}},{\mathrm{color}}_{{\left({\mathrm{seg}}_{1}*{\mathrm{seg}}_{2}\right)}^{{}_{{}^{2}}}}\right)\right]$$
where ${\mathrm{seg}}_{1}$ denotes the first cut rate and ${\mathrm{seg}}_{2}$ denotes the second slice ratio.

### 3.2. Semantic Mapping Rule Mining

Semantic mapping rules are used to express associations between underlying visual features and high-level semantics of remotely sensed images. A bag-of-words association mapping model is used to obtain this rule.

The main process is as follows: (1) Visual feature bag of words construction: Based on the feature vector set of all images in the training set, the K-means clustering algorithm is used to construct a visual feature bag of words with multiple feature vocabularies representing the basic visual features of remote sensing images, completing the mapping of features to vocabularies, which is called first mapping. (2) Association text construction: The set of words of each image is matched with the semantics (i.e., image category) of that image, and the words are recorded in the text line by line, forming a words-to-semantics association text. (3) Association rules construction: Based on the association text, the improved FP-Growth algorithm mines rules with strong association between the feature word and the image category to obtain the semantic mapping rules, which is called a two-layer mapping. When two mappings are interconnected by association text, high-level semantic information can be obtained through the underlying features shown in Figure 4.

#### 3.2.1. Visual Feature Bag-of-Words Module

The purpose of this module is to obtain words-to-semantics mining texts by con-structing mappings between features and words.

The bag of visual words [40] feature model is used in the discretization of remote sensing image data because it takes into account the similarity of spatial feature distributions when clustering features, which effectively solves the sharp boundaries problem that occurs when association rules are mined using image data directly. The following steps present the process for extracting visual feature bag of words.

Visual feature words generation

For all feature vectors, features extracted from a large amount of remote sensing image data (denoted as Rd), K-means clustering is completed using the Euclidean distance method to form *k* clustering centers, and each center corresponds to a visual feature word. After that, a dictionary $W=\left[w_{1},w_{2}\ldots w_{k}\right]$ containing visual feature words in *k* dimensions can be generated.

2.Image visual feature dictionary representation

The visual feature words can be mapped to each other and to image features by the distance algorithm, so that each small block after a secondary cut can be represented by a visual feature word, and a block after a primary cut can be described by a set of visual feature word sets, denoted as phrase.

A phrase may contain a portion of the same visual feature word, and the set obtained after de-weighting any phrase is a subset of the visual feature word *W*, and a remote sensing image will be represented by a collection of ${\mathrm{seg}}_{1}{}^{2}$ sets of words. The first slice blocks obtained from a large amount of remote sensing image data are expressed by the visual feature lexicon to form a collection (Equation (7)).
(7)$$\mathrm{Ts}=\{({\mathrm{phrase}}_{1}\to I_{1}),({\mathrm{phrase}}_{2}\to I_{2})\;....\;({\mathrm{phrase}}_{m}\to I_{m})\}$$

In Equation (7), Ts is the visual feature word collection with remote sensing image mapping expression, I is the primary cut block, and m is the product of the number of remote sensing images and ${\mathrm{seg}}_{1}^{2}$. Let the blocks in the set be represented by their corresponding class ${\mathrm{class}}_{j}$, where *j* is the class contained in this remote sensing image library. The expressions in the Ts collection are arranged sequentially to form the final words and semantic associated text.

#### 3.2.2. Association Rule Mapping Module

We use the FP-Growth algorithm to mine rule knowledge and obtain association rules for semantic mapping. However, some rules constructed by the traditional FP-Growth algorithm do not contain the meaning of deriving semantics from words. Therefore, the direct use of this algorithm is not conducive to understanding the rules that make up the remotely sensed image and increases the cost of mining.

In this study, we improve the FP-Growth algorithm for the issue that the meanings of some rules do not match the semantic derivation of remote sensing images. The basic idea is to divide the frequent item sets obtained by the association rule algorithm into two categories. Frequent item sets containing image categories defined as valid item sets v_itemset, but frequent item sets that do not contain image categories are defined as invalid item sets i_itemset.

The frequency m of each item set is extracted by traversing v_itemset, and its support support(v_itemset) is obtained by dividing the frequency m by the total number of item sets. During the rule mining process, the current item setting is extracted and the category item cg_item is removed from it to obtain the words item set lex_itemset; get the same item set as lex_itemset in i_itemset, extract its frequency n, divide the frequency n by the total number of item sets to get its support support(i_itemset), and divide support(v_itemset) by support(i_itemset) to obtain the confidence P. The process of semantic mapping rule confidence calculation is shown in Figure 5.

Let the current lex_itemset be the antecedent rule, cg_item be the posterior rule. The confidence level P denote the probability of occurrence of that rule. These constitute a valuable semantic mapping rule. All the semantic mapping rules can be obtained after the traversal is completed. The expression form of the rule is shown in (8) and (9),
(8)$$\mathrm{rule}=\left\{{\mathrm{lex\_itemset}}_{i}\to \left.{\mathrm{cg\_item}}_{i}:P_{i}\right\}\right.$$
(9)$$\mathrm{lex\_itemse}t_{i}=[{\mathrm{word}}_{i,1},{\mathrm{word}}_{i,2}\ldots {\mathrm{word}}_{i,j}]\;\{1\leq j\leq k\}$$
where i denotes the total number of generated rules and *k* is the number of words in W.

### 3.3. Rule-Based Semantic Similarity Retrieval of Remote Sensing Images

Based on the constructed visual feature bag of words and the words-semantic association rules, we can semantically represent the remote sensing images in the retrieval library. Then, based on the retrieval keywords input by the user, the similarity is calculated with the set of image semantics in the retrieval library, and the images that meet the requirements are obtained in descending order. Finally, feedback is provided for incorrect retrievals to reduce the confidence level of the incorrect semantics in the remote sensing images to improve the accuracy of the next retrieval.

#### 3.3.1. Semantic Similarity Retrieval of Remote Sensing Images

The key to accomplish words-to-semantics conversion using semantic mapping rules is that the antecedent rules need to correspond to each other with the words in the remote sensing images. First, the same feature extraction method is applied to the remote sensing images in the dataset to obtain the features. These features are then replaced with words by a visual feature dictionary. voc_set denotes the number of word sets contained in a remote sensing image,
(10)$$\mathrm{voc\_set}=[{\mathrm{voc}}_{1},{\mathrm{voc}}_{2}\ldots {\mathrm{voc}}_{m}]$$
(11)$$\mathrm{voc}=[{\mathrm{word}}_{1},{\mathrm{word}}_{2}\ldots {\mathrm{word}}_{s}]$$
where voc denotes the set of words in a block. m is the square of the first slice coefficient ${\mathrm{seg}}_{1}$. s is the square of the second slice coefficient ${\mathrm{seg}}_{2}$.

Let voc match with lex_itemset to implement the correspondence between the semantic mapping rules and remote sensing images. The matching method uses the term frequency-inverse document frequency algorithm (*TF-IDF*) [41,42] to calculate the importance of each word to the antecedent rule, as shown in Equations (12) and (13):(12)$$TF_{s,i}=\frac{n_{s,i}}{\sum_{k}n_{k,i}}$$
where $n_{s,i}$ denotes the number of occurrences of words in ${\mathrm{lex\_itemset}}_{i}$, and the denominator denotes the number of all words in the ${\mathrm{lex\_itemset}}_{i}$,
(13)$${\mathrm{IDF}}_{s}=\log^{\frac{N}{n+1}}$$
where *N* denotes the total number of rules contained in the antecedent rule, and *n* denotes the number of rules containing ${\mathrm{word}}_{s}$. The denominator plus 1 is to prevent the case that the word is not in the rule, resulting in a denominator of 0. The *IDF* value indicates that if fewer rules contain ${\mathrm{word}}_{s}$, the ${\mathrm{word}}_{s}$ have a better ability to distinguish the semantics of remote sensing images. From Equations (12) and (13), we obtain the *TF-IDF* value using Equation (14), which indicates the importance of ${\mathrm{word}}_{s}$ in a rule.
(14)$$\mathrm{TF}-{\mathrm{IDF}}_{s}={\mathrm{TF}}_{s,i}\times {\mathrm{IDF}}_{s}$$

After obtaining *TF-IDF* values of all words in voc, the semantic probabilities obtained in the semantic mapping rules are used as added weights to jointly calculate the degree of matching with Equation (15):(15)$$\mathrm{mate}({\mathrm{voc}}_{m}{}^{\wedge }{\mathrm{lex\_itemset}}_{i})=\mathrm{rule}(P)*\sum_{k=1}^{s}{\mathrm{TF}}_{k,i}\times {\mathrm{IDF}}_{k}$$
where m denotes the number of word sets, i denotes the number of antecedent rules, rule(P) denotes the confidence of the current matching rule, and s denotes the number of words in the current matched word sets. The rule with the highest matching degree with the word set is taken. To complete the conversion of all the word sets in each image to the semantic set, the semantic information and confidence degree in the rule are used to represent the word set. Each semantic meaning in the semantic set has a corresponding confidence degree, which indicates the probability of the occurrence of that semantic meaning.

According to Equation (16), the cosine distance between the retrieved keywords and the semantic set of each image in the remote sensing image retrieval library is calculated and the results are detected between 0 and 1 and the output in descending order. As the value of the result is closer to 1, the more similar it is to the semantics of the retrieved word:(16)$$\cos\theta =\frac{\sum_{i=1}^{n}\left(A_{i}\times B_{i}\right)}{\sqrt{\sum_{i=1}^{n}A_{i}^{2}\times \sum_{i=1}^{n}B_{i}^{2}}}$$
where $A_{i}$ denotes the semantic probability of the retrieved keyword, $B_{i}$ denotes the semantic probability of the images, and n denotes the number of semantic.

#### 3.3.2. Error Retrieval Feedback

Incorrect images may appear in the retrieval results. The reason for retrieving an incorrect image is that the semantic set of that image has a certain degree of similarity with the retrieved keywords. Therefore, in order to improve the retrieval accuracy, we designed a feedback mechanism whose main idea is to reduce the probability of the retrieval keyword appearing in the wrong image. The process is illustrated in Figure 6.

Remote sensing image retrieval results have m error images, and the set of error semantic in each error image is denoted as *s_set*. The semantic *s*, and its corresponding probability P, are included in the set of error semantics. As shown in Equation (17),
(17)$${\mathrm{s\_set}}_{m}=[s_{1}:P_{1},s_{2}:P_{2}\ldots s_{n}:P_{n}]$$
where *n* denotes the number of semantics contained in the semantic set.

The semantic items in the semantic set are traversed, and the items that are semantically similar to the retrieved keyword are filtered out. A correction number cor is set to reduce the semantic probability.

Traverse *s_set_m_* to filter out semantic items that are semantically similar to the retrieved keyword. cor is set to reduce the semantic probability in these items. The reduced probability is at least 0, as shown in Equation (18).
(18)$${\mathrm{cor\_set}}_{i}=\left\{\begin{array}{l} s_{i}:P-\mathrm{cor}\;s_{i}=\mathrm{keyword}\\ s_{i}:P\;s_{i}\neq \mathrm{keyword} \end{array}\right.\;i=\{1,2\ldots n\}$$

All the semantic items in *s_set_m_* are substituted into Equation (18) to correct the semantic to obtain the new semantic set. The new semantic set is used for semantic similarity calculation during the re-retrieving. As only the probability of incorrect semantic in an incorrect image is reduced, it does not affect the semantic similarity calculation.

## 4. Experiment and Analyses

This section provides a concise and precise description of the experimental results, their interpretation, and the experimental conclusions that can be drawn.

### 4.1. Details of Implementation

#### 4.1.1. Experimental Setup

Hardware platform for the experiments: Intel Core i7-7700K CPU@4.2 GHz, 32 G RAM, 256 G SSD, Ho Chi Minh City, Vietnam. The programming language is python 3.6.

The experimental software environment include the following: operating system Windows 10, PyCharm 2020.3, Torch version 1.3.1, Torch vision version 0.4.1.

#### 4.1.2. Dataset

Aerial Image Dataset (AID) dataset [34]

This dataset contains multiple scene categories acquired from Google Earth using multiple sensors. It is divided into 30 scene categories, for a total of 10,000 images. The number of images in each category varies from 220 to 420, and the categories include airports, bare land, centers, churches, businesses, dense housing, and deserts.

2.NWPU-RESISC45 dataset [36]

The dataset contains 31,500 remote sensing images with 45 scene classes such as aircraft, airports, baseball fields, basketball courts, beaches, bridges, and clouds. Each class contains 700 image patches.

#### 4.1.3. Evaluation Indicators

In this study, the experiments use the mean average precision, average normalized modified retrieval rank, and retrieval time as measures of the performance of the image retrieval system.

Mean average precision (*mAP*)

In a remote sensing image to be queried and a retrieved image dataset with the number of images *R*, the value of average precision (*AP*) is calculated as defined below:(19)$$\mathrm{AP}=\frac{1}{r}\sum_{k=1}^{R}\mathrm{correct\_num}(k)/k$$
where *r* denotes the number of images of the same category as the image to be retrieved in the image dataset, and *k* denotes the search position in the query result. correct_num(k) denotes the number of images of the same category as the image to be retrieved returned to the *k*th retrieved image in the image query sequence. correct_num(k) is 0 if the *k*th image does not match the category of the image to be retrieved. Averaging the *AP* values for all queries yields *mAP* (Equation (20)):(20)$$\mathrm{mAP}=\frac{\sum_{n=1}^{N}\mathrm{AP}(n)}{N}$$
where *N* denotes the number of retrievals. *mAP* takes values in the range of [0, 1], and the larger the value, the higher the retrieval accuracy.

2.Average normalized modified retrieval rank (*ANMRR*)

The images in the remote sensing image retrieval dataset all have *Rank(k)*, indicating their corresponding ranking positions. The semantic retrieval keyword performed each time is *SK*. The set of images related to the semantic of the retrieved word exists in the remote sensing image retrieval dataset, the number of which is *B(SK)*. In the retrieval list, the first *K* images are taken. If *Rank(k)* is greater than *K*, *Rank(k)* is set to 1.25 × *K*, as shown in Equation (21).
(21)$$\mathrm{Rank}(k)=\left\{\begin{array}{l} \mathrm{Rank}(k)\;\mathrm{Rank}(k)\leq K\\ 1.25\times K\;\mathrm{Rank}(k)>K \end{array}\right.$$

The average rank AVR(SK) of the query *SK* is denoted by:(22)$$\mathrm{AVR}(\mathrm{SK})=\frac{1}{B(\mathrm{SK})}\sum_{k=1}^{B(\mathrm{SK})}\mathrm{Rank}(k)$$

The normalized corrected retrieval rank is set to NMRR(SK):(23)$$\mathrm{NMRR}(\mathrm{SK})=\frac{\mathrm{AVR}(\mathrm{SK})-0.5(1+B(\mathrm{SK}))}{1.25K-0.5(1+B(\mathrm{SK}))}$$

When the performance of the image retrieval method is evaluated in the remote sensing image set, *M* queries are performed, and we leverage ANMRR i.e., a specific evaluation criterion for image retrieval (Equation (24)) to access the retrieval performance.
(24)$$\mathrm{ANMRR}=\frac{1}{M}\sum_{\mathrm{SK}=1}^{M}\mathrm{NMRR}(\mathrm{SK})$$

ANMRR is a very specific evaluation criterion for image retrieval. If all the relevant images are found in the retrieved image, then the value of ANMRR is 0, otherwise it is 1. Therefore, a lower ANMRR indicates better retrieval performance.

3.Retrieval time

Retrieval time is taken as one of the evaluation indexes. It includes the processes from the input of retrieval keywords to get the sequence of retrieval results.

### 4.2. Experimental Procedure

Any 20 different types of remote sensing images are arbitrarily selected from the dataset of which 70% are used as remote sensing image data for training semantic mapping rules and 30% are used as a remote sensing image retrieval dataset for checking the image retrieval performance. Set both slice ratios to 4 × 4 in the feature extraction module.

#### 4.2.1. Semantic Mapping Rule Mining Experiment

To obtain semantic mapping rules required for the semantic retrieval, we need to set some parameters when mining the rules. Since the feature word is obtained from the training data using the K-means clustering center algorithm, an optimal clustering value is used to ensure that different remote sensing image features can be distinguished as much as possible, so the credibility of the information derived from the rules is improved. The method uses a distance cost-effectiveness function with the following equation:(25)F(S,k)=L+D
(26)$$L=\sum_{i=1}^{k}\left|m_{i}-m\right|$$
where *L* denotes the interclass distance, *m* is the mean of all samples, *m_i_* is the mean of the samples contained in the clusters, and *k* is the number of all clusters:(27)$$D=\sum_{i=1}^{k}\sum_{p\in C_{i}}\left|p-m_{i}\right|$$
where *D* is the intra-class distance, *p* is a sample, *m_i_* is the mean of the samples contained in the clusters, and *k* is the number of clusters to be clustered. The corresponding *k* value is the best feature word when the value of *F(S,k)* is the smallest.

A visual feature dictionary is created using the optimal number of characteristics as a base. The Euclidean distance is used to label the remote sensing image features in the training set with the visual feature dictionary to obtain the words-to-semantics association text. The words-to-semantics association texts are substituted into the improved Fp-growth algorithm to obtain the semantic mapping rules. The performance of the improved Fp-growth algorithm is evaluated in terms of the mining time, the number of rules mined, and the efficiency of the rules. For comparison, 500, 600, 700, 800, 900, and 1000 random remote sensing images in the AID dataset and NWPU-RESISC45 dataset are selected for comparison.

#### 4.2.2. Remote Sensing Image Semantic Retrieval Performance Evaluation

For the performance evaluation of remote sensing image retrieval, the proposed method is compared with the VGG16 and ResNet18 by retrieving *mAP* of different categories in the AID and NWPU-RESISC45.

Since the VGG16 model and ResNet18 model need to complete semantic retrieval in order to compare with our methods, the adopted strategy is used to complete category annotation for images in the remote sensing image retrieval dataset and this is used to achieve semantic retrieval. The method used in this study is then compared with VGG16, VGG19, ResNet18, ResNet34, and the method in the literature [23] in terms of ANMRR values and retrieval time to demonstrate the superiority of the semantic retrieval of remote sensing images achieved by this method in terms of retrieval performance.

#### 4.2.3. Performance Evaluation of Remote Sensing Image Feedback Mechanism

For the performance evaluation of the remote sensing image feedback mechanism proposed in this paper, five categories with low *mAP* in the AID dataset were selected for multiple feedbacks, and the changes in the *mAP* of each category after each feedback were observed.

### 4.3. Experimental Results and Analysis

In this section, three main experiments are conducted, all of which are unfolded with a confidence level of 0.6 and a support level of 0.4 settings.

#### 4.3.1. Semantic Mapping Rule Mining Results

Firstly, using the distance cost-effectiveness function, as shown in (25)–(27), the starting number of feature words is set to 2, and by selecting a different number of feature words in ascending sequence. The values of the feature sets derived from the two datasets of AID and NWPU-RESISC45 are calculated. The calculation results are shown in Figure 7.

As can be seen from Figure 7, the *F(S,k)* values in the feature vocabulary obtained for dataset AID and dataset NWPU-RESISC45 tend to stabilize at *k* = 80 and *k* = 54. Therefore, according to the distance cost minimization criterion, *k* = 80 and *k* = 54 are chosen as the optimal number of feature words for dataset AID and dataset NWPU-RESISC45.

In this study, the FP-Growth algorithm is optimized to obtain valuable semantic mapping rules. The reliability of the performance of the optimization algorithm can be seen by comparing the mining time, the number of association rules established, and the percentage of valid association rules with the traditional methods.

From the comparison experiments in Figure 8a,b, it can be seen that the optimized Fp-growth algorithm spends significantly less time on rule mining than the traditional algorithm. Figure 9a,b show that the optimized algorithm eliminates a large number of invalid rules compared to the traditional algorithm. Figure 10a,b show that the effective rule generation rate of the optimized algorithm is much higher than that of the traditional algorithm. As the number of images to be mined increases, the traditional algorithm increases excessively in terms of mining time and number of rules, while the optimized algorithm is more stable. Therefore, the optimized algorithm is significantly better than the traditional algorithm in the overall view.

The impact of the Fp-growth algorithm on retrieval performance before and after optimization is verified in two datasets, and the results are listed in the following table.

It can be seen from Table 1 that the overall performance of remote sensing image retrieval using the optimized FP-growth algorithm is better than the retrieval performance without the optimized algorithm. In the comparison experiments between the two datasets *mAP* improved by 16.8% and 10.9%, ANMMR values decreased by 0.1649 and 0.1173, and time decreased by 0.0892 s and 0.671 s. This is because the mined rules containing invalid rules do interfere with the semantic shape calculation performed in remote sensing image retrieval, leading to decrease in retrieval accuracy. A larger number of invalid rules enhance semantic matching time, and thus make retrieval time growth.

#### 4.3.2. Performance Evaluation of Semantic Retrieval of Remote Sensing Images

The three methods were used to semantically assign the remote sensing images in the AID and NWPU-RESISC45 dataset, respectively, and the retrieval results are obtained using the retrieval keywords. The *mAP* of each category were calculated in the two datasets for comparison, as shown in Figure 11.

As shown in Figure 11a, the retrieval accuracy achieved by our method is higher, with the average *mAP* of 87.5%, improving by 22.5% and 17.6% compared to the other two methods, respectively. However, some of these categories—such as central area, parking lot, and pond—have poor retrieval accuracy. An example of a retrieval of these categories is shown in Figure 12. The reason for the low retrieval accuracy of the central area, parking, and pond categories from the analysis in Figure 12 may be that some images in these categories contain a large number of elements from other categories, resulting in inaccurate matching of the features with semantics.

Similar results are shown in Figure 11b, where the average *mAP* of the dataset NWPU-RESISC45 reaches 90.8%, improving 20.3% and 15.6% compared to the other two methods. The retrieval accuracy of image retrieval in the dataset NWPU-RESISC45 is higher than that in the AID dataset. Due to the relatively small image resolution of the NWPU-RESISC45 dataset, which focuses more on the core features described by the image semantic, the features extracted from it can play a better role in semantic differentiation. This leads to a more consistent derivation of each rule with the actual semantic in the process of constructing rules, and thus improving the retrieval accuracy.

The comparison between the method in this paper and other methods in terms of ANMRR values versus time is listed in Table 2 and Table 3.

As listed in Table 2 and Table 3, the method in this study improves the retrieval accuracy and reduces the retrieval time in terms of ANMRR versus time. We selected VGG19, ResNet34, Literature [23] and our method to perform semantic retrieval in the AID dataset. The top 5 images are returned for retrieval as an example as shown in Figure 13.

The retrieval examples shown in Figure 14 are the results returned by semantic retrieval using VGG19, ResNet34, Literature [23] and our method on the dataset NWPU-RESISC45.

#### 4.3.3. Feedback Mechanism Effect Evaluation

Five categories with low average inspection rates were selected from the retrieval results of the AID dataset: center, meadow, medium, residential, park, and pond. The results are shown in Figure 15.

This experiment shows the retrieval results after seven feedbacks, setting the correction value cor to 0.1. 0 in the horizontal coordinate indicates the initial *mAP* without feedback.

The first two feedbacks have the largest improvement in the *mAP*—up to 12%, while the average value reaches 7.02%. The last three feedback results showed an increase, but the increase was relatively insignificant. From the overall trend, the feedback mechanism designed in this paper can effectively correct the semantic probability of error retrieval images and thus improve the image semantic retrieval accuracy. Although the effect of this mechanism on the *mAP* improvement of each category appears to decrease significantly with the increase in the number of feedbacks, better retrieval results can always be obtained after multiple feedbacks.

## 5. Discussion

To address the semantic gap between the underlying features and high-level semantic of remote sensing images, the association rule algorithm in data mining technology is used to mine a large amount of remote sensing image data to obtain the semantic mapping rules for deriving high-level semantics of remote sensing images. The semantic mapping rules can automatically assign semantic sets to remote sensing images and complete the semantic retrieval of remote sensing images through semantic sets. Two slices performed on remote sensing images, setting the support and confidence in semantic mapping rule mining, and the semantic probability are important parameters that affect the semantic retrieval of remote sensing images by the method. Our method provides an easy-to-implement idea for semantic-based remote sensing image retrieval.

### 5.1. Effect of Slice Ratio on Remote Sensing Image Retrieval

The purpose of setting the slice rate is to extract the small targets from remote sensing images. When the targets representing the dominant semantics of a remote sensing image are extracted as much as possible, then the set of semantics derived for that image is more biased towards its dominant semantics. The first slice determines the number of semantics describing the image, and the second slice determines the number of words describing the semantics. The slice ratio is set for both slices, and experiments are conducted in dataset AID to discuss the effect of the slice ratio on the images retrieved from remote sensing images.

As can be seen from Table 4, the retrieval accuracy substantially increases when the number of semantics describing the image increases. The retrieval accuracy also somewhat improves as the number of words used to describe the semantic increases. This indicates that extracting the small targets in the image by slicing the image is significantly helpful in deriving the dominant semantic of the image, and the small targets in the image are described more carefully as the proportion of slices increases. This reduces the possibility of inferring the wrong dominant semantics and thus improvs the accuracy of retrieval. In addition, the increase in the number of semantics and words leads to a sacrifice in retrieval time. However, the comparison between the data with the first slice ratio of 5 × 5 and the data with the first slice ratio of 4 × 4 shows that the image retrieval accuracy decreases with the slice value rising. Thus, it is important to set the appropriate slice ratio for the dominant semantic distribution pattern of remote sensing images.

### 5.2. Impact of Support and Confidence on Remote Sensing Image Retrieval

The core of this work is to construct semantic mapping rules. The connection between the underlying visual features and high-level semantics of remote sensing images is established by an improved FP-Growth algorithm.

Two important pre-set parameters are used in mining the rules: support and confidence. The setting of support affects the composition of frequent item sets and confidence affects the composition of semantic mapping rules. Therefore, different support and confidence levels are set to the impact on the accuracy of remote sensing image retrieval using the AID dataset for mining, so that both slice ratios are 4 × 4, as listed in Table 5, Table 6 and Table 7.

Table 5, Table 6 and Table 7 indicate that increasing the set support and confidence values results in a considerable drop in the number of frequent item sets, the number of rules, and the mining time. The *mAP* used to measure retrieval accuracy does not improve due to the increase in rule probability. The possible reason is that the increase in confidence level results in a significant reduction in the derived rules.

Analysis of the support in each table gives similar results. Lower or higher support levels lead to a decrease in retrieval accuracy, and the relatively better results are found in the middle interval between 0 and 1. This work selects support and confidence from the middle interval, so good retrieval accuracy is always obtained. However, the optimal support and confidence cannot be pinpointed.

It is noted that the retrieval time of the data sets with low retrieval precision is generally slightly higher than that of the data sets with high retrieval precision. This may be due to the obvious difference in semantics, leading to the need for multiple matches during retrieval, thus causing the retrieval time.

### 5.3. Influence of Semantic Probability as Additive Weights on Remote Sensing Image Retrieval

This work realizes the matching of word sets in retrieved images and rule sets in semantic mapping rules by Equation (15) and completes the semantic assignment of remote sensing images according to the matching degree. This work measures the matching degree based on the use of *TF-IDF* values together with the rule probability. The more accurate the matching is, the higher the subsequent retrieval accuracy is. Therefore, the effect of adding semantic probability on matching is evaluated by retrieval accuracy in both datasets, as listed in Table 8 and Table 9.

Table 8 and Table 9 show that the matches determined using the *TF-IDF* values result in a lower accuracy in remote sensing image retrieval. Only the similarity between the set of rule words and the set of image words is considered to make the result that some rules with low semantic probability match better than those with high semantic probability as matching semantics. This situation generally does not affect the retrieval accuracy. It is contrary to the fact that the higher the semantic probability, the more realistic the resulting rule is. However, a balance between phrase similarity and semantic probability of each rule can be achieved when employing semantic probability along with *TF-IDF* values to calculate the match, increasing the accuracy of image retrieval. This also implies that the semantic probability is an important parameter for improving the matching degree.

## 6. Conclusions

In this paper, we propose a bag-of-words association mapping method by which the derived semantic mapping rules can visually represent the derivation process from features to semantics as an effective solution to the semantic gap problem in image retrieval. A comparison with different models in two datasets leads to the following conclusions:(1)The optimized FP-growth algorithm can effectively improve the speed of rule mining by eliminating a large number of invalid rules and making the obtained rule set consist of valid rules as much as possible. The algorithm eliminates the interference by invalid rules in subsequent remote sensing image retrieval, thus improving the accuracy of remote sensing image retrieval.(2)The semantic retrieval of remote sensing images realized by the method used in this study is higher in overall retrieval accuracy than the traditional, and has significantly shortened the retrieval time. This is because the semantic mapping rules obtained with association rule mining on a large amount of image data can construct a strong association between the underlying visual features and high-level semantics.(3)Correction of the semantic probability of errors in incorrectly retrieved images using the feedback mechanism can effectively improve the accuracy of the next retrieval of similar semantic images. In particular, the accuracy will be improved substantially after multiple feedback loops.

For remote sensing images with rich spatial information, the semantic information contained in them is not independent and can often be described by multiple semantics together. Therefore, it is necessary to integrate the multidimensional semantics of remote sensing images with the rules when constructing the rules and it is also necessary to retain more semantic information in the extracted remote sensing image features. In addition, support for and confidence in the proposed method are important parameters that affect rule generation. If the parameter is set too low, it will lead to the appearance of rules with low confidence, and conversely, it will lead to the deletion of key rules that may be needed. Then, it is necessary to find an optimal set of support and confidence to get the best retrieval accuracy possible. This will be the main focus of the next research.

## Figures and Tables

**Figure 1 sensors-23-05807-f001:**
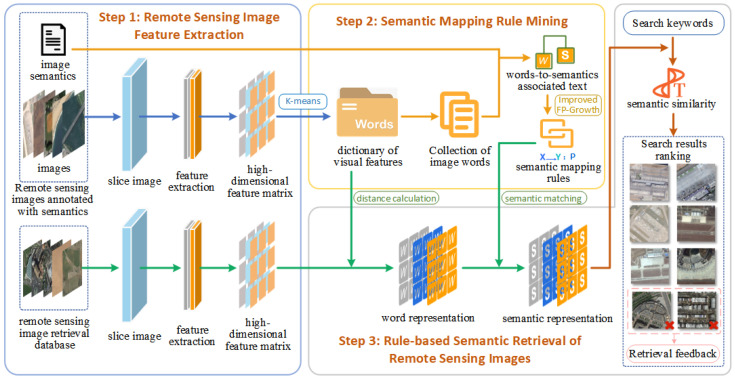
Semantic retrieval of remote sensing images based on the bag-of-words association mapping method.

**Figure 2 sensors-23-05807-f002:**
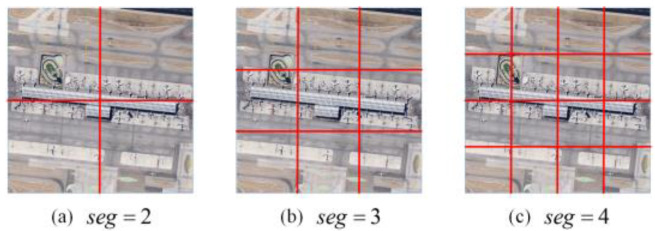
Seg of the image slice (The value of seg means that an image is going to be split equally into seg × seg blocks): (**a**) The value of seg is 2 and the image is split equally into 4 pieces. (**b**) The value of seg is 3 and the image is split equally into 9 pieces. (**c**) The value of seg is 4 and the image is split equally into 16 pieces.

**Figure 3 sensors-23-05807-f003:**
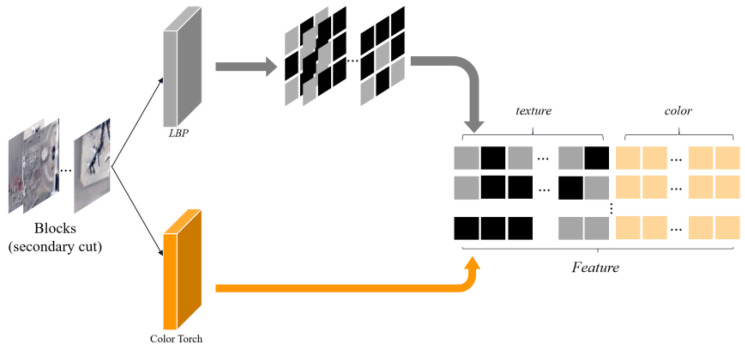
Texture and color feature vector extraction.

**Figure 4 sensors-23-05807-f004:**
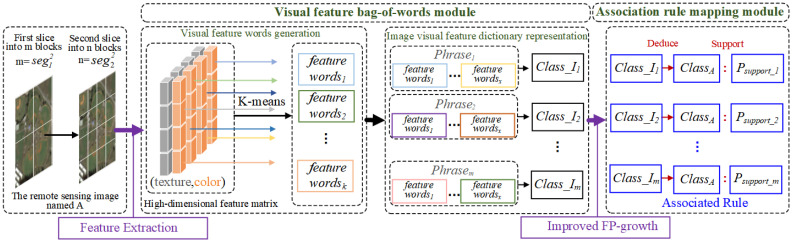
Bag-of-words association mapping method.

**Figure 5 sensors-23-05807-f005:**
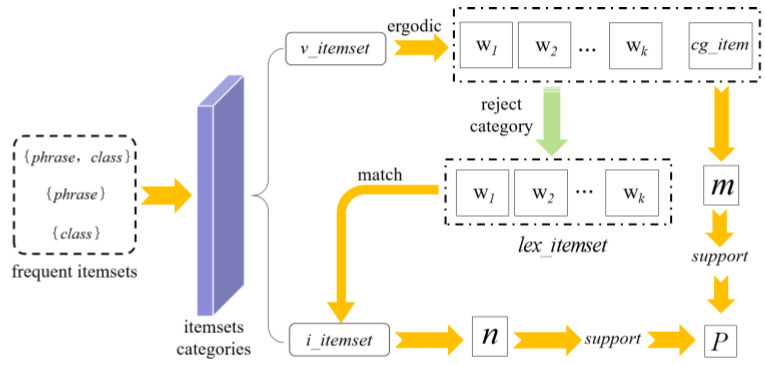
Flow chart of semantic mapping rule confidence calculation.

**Figure 6 sensors-23-05807-f006:**
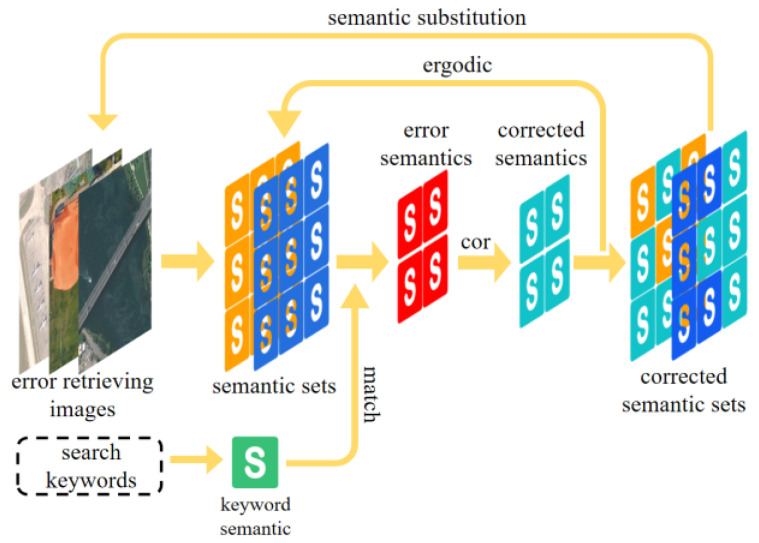
The feedback mechanism.

**Figure 7 sensors-23-05807-f007:**
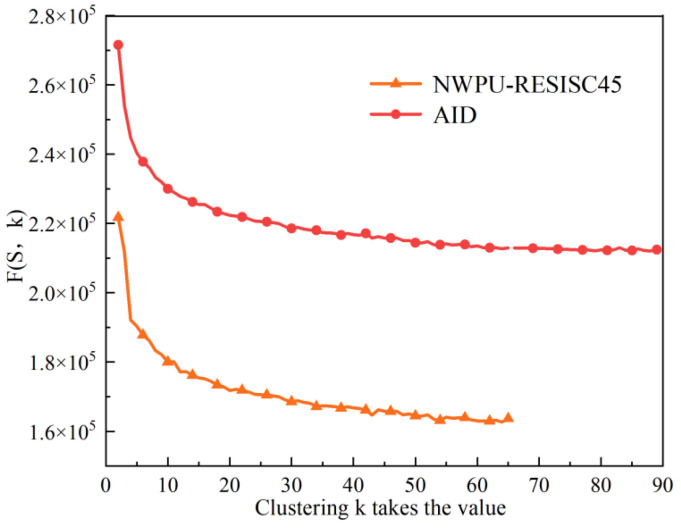
Results of feature word calculation for different data sets.

**Figure 8 sensors-23-05807-f008:**
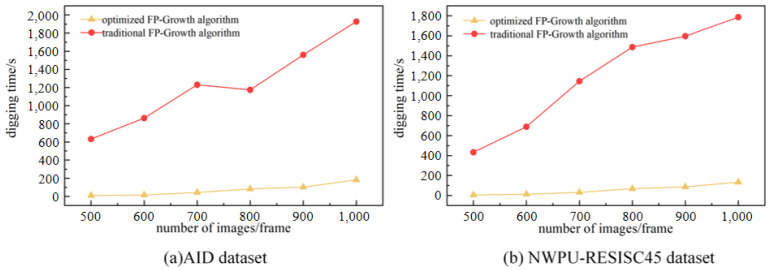
Semantic mapping rule mining time comparison: (**a**) Digging time on the AID dataset (**b**) Digging time on the NWPU-RESISC45 dataset.

**Figure 9 sensors-23-05807-f009:**
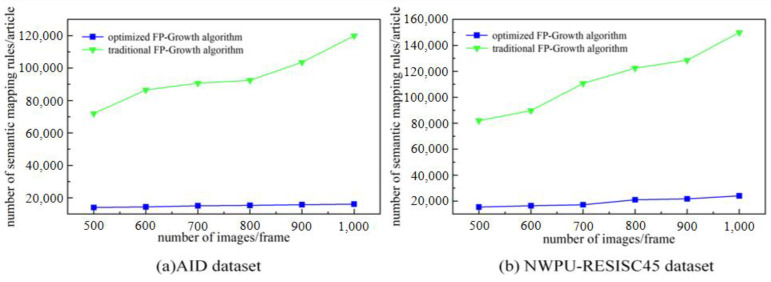
Comparison of the number of semantic mapping rules mining rules: (**a**) Number of semantic mapping rules mined on the AID dataset (**b**) Number of semantic mapping rules mined on the NWPU-RESISC45 dataset.

**Figure 10 sensors-23-05807-f010:**
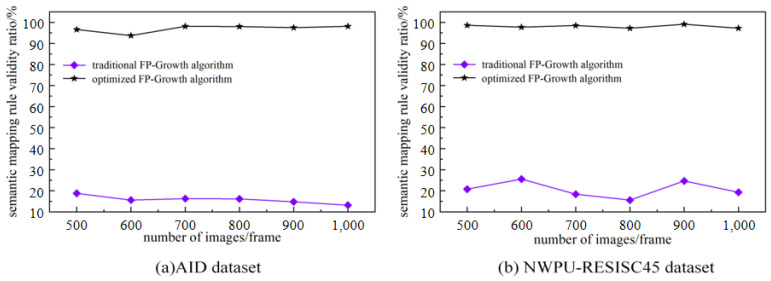
Comparison of effective rule generation ratio of semantic mapping rules: (**a**) Semantic mapping rule validity on the AID dataset (**b**) Semantic mapping rule validity on the NWPU-RESISC45 dataset.

**Figure 11 sensors-23-05807-f011:**
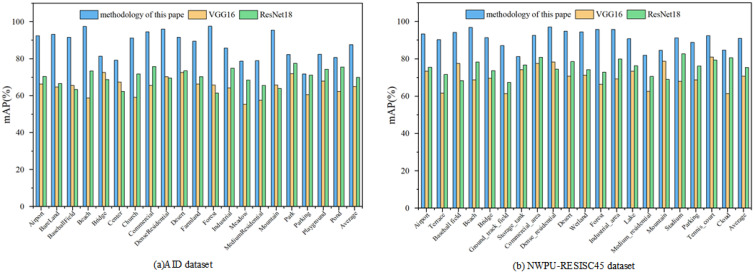
Comparison of the retrieval result category *mAP* between the method of this paper and other methods: (**a**) Retrieval result category *mAP* for the three methods on the AID dataset (**b**) Retrieval result category *mAP* for the three methods on the NWPU-RESISC45 dataset.

**Figure 12 sensors-23-05807-f012:**
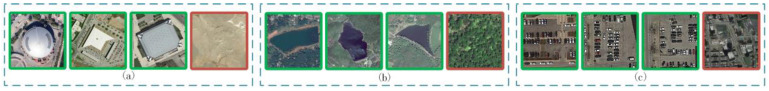
Error images when retrieving the central area, parking, and pond, categories (Red indicates error image): (**a**) Retrieval image of central area. (**b**) Retrieved image of pond. (**c**) Retrieved image of parking.

**Figure 13 sensors-23-05807-f013:**
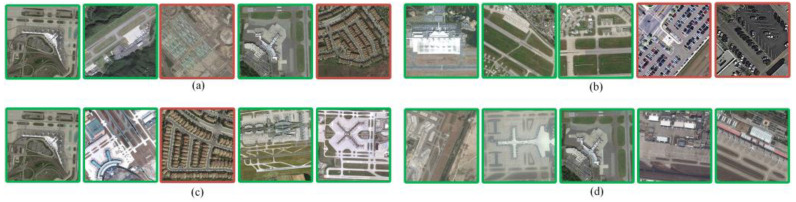
Retrieval results using airport as a retrieval word in the AID dataset (Green indicates correct retrieval results, red indicates incorrect retrieval results): (**a**) Images retrieved using VGG19. (**b**) Images retrieved using ResNet34. (**c**) Images retrieved using the Literature [23] method. (**d**) Images retrieved using the methods in this paper.

**Figure 14 sensors-23-05807-f014:**
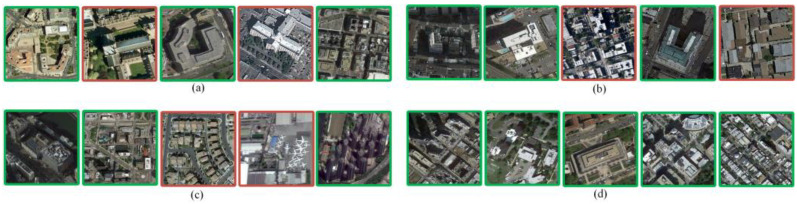
Retrieval results using commercial areas as a retrieval term in the NWPU-RESISC45 dataset (Green indicates correct retrieval results, red indicates incorrect retrieval results): (**a**) Images retrieved using VGG19. (**b**) Images retrieved using ResNet34. (**c**) Images retrieved using the Literature [23] method. (**d**) Images retrieved using the methods in this paper.

**Figure 15 sensors-23-05807-f015:**
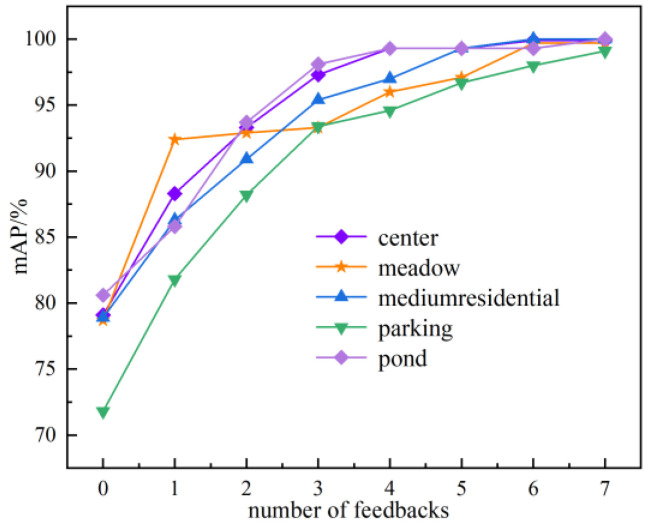
Effect of feedback mechanism on image retrieval accuracy.

**Table 1 sensors-23-05807-t001:** Effect of Fp-growth algorithm on retrieval performance before and after optimization.

Method	Dataset	mAP/%	ANMMR	Time/s
optimized FP-Growth algorithm	AID	87.5	0.0779	0.0648
	NWPU-RESISC45	90.8	0.0695	0.0256
traditional FP-Growth algorithm	AID	70.7	0.2428	0.154
	NWPU-RESISC45	79.9	0.1868	0.0927

**Table 2 sensors-23-05807-t002:** ANMRR versus time for this paper’s method versus other methods in dataset AID.

Method	ANMRR	Time/s
VGG16	0.3012	0.2896
VGG19	0.2897	0.2758
ResNet18	0.2512	0.1598
ResNet34	0.2453	0.1655
Literature [23]	0.2063	0.1165
methodology of this paper	0.0779	0.0648

**Table 3 sensors-23-05807-t003:** Comparison of ANMRR and time of this paper’s method with other methods in the dataset NWPU-RESISC45.

Method	ANMRR	Time/s
VGG16	0.2495	0.1631
VGG19	0.2343	0.1597
ResNet18	0.2218	0.1396
ResNet34	0.1987	0.1327
Literature [23]	0.1755	0.1055
methodology of this paper	0.0695	0.0256

**Table 4 sensors-23-05807-t004:** Effect of slicing ratio on image retrieval performance.

Slitting Ratio		mAP/%	ANMRR	Time/s
First Slice	Second Slice			
2 × 2	2 × 2	70.5	0.2503	0.0293
2 × 2	3 × 3	73.1	0.2276	0.0317
2 × 2	4 × 4	74.0	0.2198	0.0388
3 × 3	2 × 2	75.8	0.2015	0.0371
3 × 3	3 × 3	81.4	0.1466	0.0402
3 × 3	4 × 4	83.4	0.1307	0.0477
4 × 4	2 × 2	77.2	0.1832	0.0408
4 × 4	3 × 3	84.3	0.1013	0.0561
4 × 4	4 × 4	87.5	0.0779	0.0648
5 × 5	2 × 2	79.3	0.1678	0.0576
5 × 5	3 × 3	84.7	0.1124	0.0975
5 × 5	4 × 4	82.6	0.1410	0.1196

**Table 5 sensors-23-05807-t005:** Effect of support on image retrieval performance at confidence level of 0.5.

Support	Number of Frequent Item Sets	Number of Rules	Digging Time/s	mAP/%	Retrieval Time/s
0.3	95,437	36,874	672.114	71.9	0.0918
0.4	71,892	22,587	371.633	80.1	0.0828
0.5	68,741	20,536	244.887	82.9	0.0702
0.6	61,054	19,883	205.853	83.7	0.0762
0.7	39,875	13,381	163.870	69.0	0.0948
0.8	11,658	9892	127.302	55.9	0.1081

**Table 6 sensors-23-05807-t006:** Effect of support on image retrieval performance at confidence level of 0.6.

Support	Number of Frequent Item Sets	Number of Rules	Digging Time/s	mAP/%	Retrieval Time/s
0.3	95,437	20,476	487.658	74.6	0.0934
0.4	71,892	16,095	182.347	87.5	0.0648
0.5	68,741	15,569	163.985	82.3	0.0603
0.6	61,054	12,458	103.651	85.6	0.0725
0.7	39,875	7654	65.578	66.8	0.1068
0.8	11,658	4787	36.885	53.3	0.1157

**Table 7 sensors-23-05807-t007:** Effect of support on image retrieval performance at confidence level of 0.7.

Support	Number of Frequent Item Sets	Number of Rules	Digging Time/s	mAP/%	Retrieval Time/s
0.3	95,437	15,308	254.101	69.3	0.1147
0.4	71,892	11,041	112.544	77.1	0.0907
0.5	68,741	9045	96.750	74.7	0.0896
0.6	61,054	8712	90.047	63.8	0.1071
0.7	39,875	4835	40.310	58.0	0.1136
0.8	11,658	1887	24.607	43.4	0.1289

**Table 8 sensors-23-05807-t008:** Effect of rule probability on image retrieval performance (AID dataset).

Method	mAP/%	ANMRR
TF-IDF	79.6	0.1613
TF-IDF + rule(P)	87.5	0.0779

**Table 9 sensors-23-05807-t009:** Effect of rule probability on image retrieval performance (NWPU-RESISC45 dataset).

Method	mAP/%	ANMRR
TF-IDF	86.1	0.0874
TF-IDF + rule(P)	90.8	0.0695

## Data Availability

All data are open public data and able to be downloaded free of charge. The Aerial Image Dataset (AID) is available at https://paperswithcode.com/dataset/aid accessed on 11 January 2023. The NWPU-RESISC45 dataset is available at https://1drv.ms/u/s!AmgKYzARBl5ca3HNaHIlzp_IXjs accessed on 16 January 2023.

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
