# Peer review of "Semantic Retrieval of Remote Sensing Images Based on the Bag-of-Words Association Mapping Method"

_sensors, 2023, doi:10.3390/s23135807_

Round 1

Reviewer 1 Report

I examined your work titled “Semantic retrieval of remote sensing images based on bag-of-words association mapping method” in detail. I would like to point out that the work is well written. However, some points need to be reconsidered by researchers. The researchers proposed a hybrid model in their study. In the study, a word bag association mapping method using two mappings from features to words and from words to semantics is proposed to improve the interpretability of the semantic-based cross-modal retrieval process. There are many typographical errors in the study. Particular attention was paid to uppercase and lowercase letters. Example 3.2.2. words-to-semantics association rule construction. In the abstract, the answer to why the study was done should be given. Also, would the results be lower when direct classification methods were applied to these datasets, for example? Classification studies using relevant data sets in the literature can be added to the literature review section. Figure 4 may be difficult to understand, the relevant figure should be detailed. Row 570 will be Table 2, not Table 1, I guess. Limitations of the study should be addressed. We, our style should be avoided as much as possible. I believe that the work will come to a better point after the revision.

The study should be reviewed.

Reviewer 2 Report

1: it is a large manuscript, could it be reduced?  

2: I was wondering if in the comparison experiments between the two datasets mAP values improved by 16.8% and 7.9%,  (line 573) and in  the average mAP 20.3 % (line 606) can have a confident interval

Reviewer 3 Report

The paper lacks concrete examples of classification for images from various fields to highlight the efficiency of the proposed algorithm.
